# Association between blood lipid levels and risk of gastric cancer: A systematic review and meta-analysis

**Shicong Xu, Ying Fan, Yuyue Tan, Ling Zhang, Xianrong Li**  *

Department of Gastrointestinal surgery, The Affiliated Hospital of Southwest Medical University, Luzhou, China

* 1551894592@qq.com

## Abstract

### Objective

The association between blood lipid levels and the risk of gastric cancer (GC) is well known. Therefore, to clarify this association, all relevant prospective cohort studies were included in this meta-analysis.

### Methods

Our study was registered in PROSPERO (CRD42022354899) prior to its commencement. A systematic review and meta-analysis were conducted in accordance with the PRISMA recommendations. Chinese databases (CNKI, CBM, Wanfang, and VIP) and English databases (PubMed, Embase, Web of Science, and the Cochrane Library) were systematically searched up to October 2022. This study included all relevant cohort studies that reported hazard ratios (HRs) or relative risks (RRs) and their corresponding 95% confidence intervals (95% CIs) to examine the association between various lipid profiles (e.g., total cholesterol, triglycerides, high-density lipoprotein cholesterol, and low-density lipoprotein cholesterol) and the risk of developing gastric cancer (GC). Fixed effects or random effects models were used based on the level of heterogeneity among the studies, and these models were employed to obtain pooled hazard ratios. Additionally, sensitivity analysis and publication bias analysis were conducted to ensure the robustness and reliability of the findings.

### Results

After conducting a systematic search, a total of 10 studies were selected out of 10,525 papers involving a total of 5,564,520 individuals. Among these individuals, there were 41,408 GC cases. The analysis revealed that the highest versus lowest serum total cholesterol (TC) concentration was associated with a pooled hazard ratio of 0.89 (95% CI = 0.87–0.92, $I^2$ = 15%). For triglycerides (TGs), the hazard ratio was 1.00 (95% CI = 0.96–1.04, $I^2$ = 37%), while for high-density lipoprotein cholesterol (HDL-C), the hazard ratio was 0.90 (95% CI = 0.86–0.93, $I^2$ = 0%). The hazard ratio for low-density lipoprotein cholesterol (LDL-C) was 0.96 (95% CI = 0.91–1.00, $I^2$ = 0%).

**Data Availability Statement:** All relevant data are within the paper and its Supporting information files.

**Funding:** The author(s) received no specific funding for this work.

**Competing interests:** The authors have declared that no competing interests exist.

## Conclusions

Based on the results of this meta-analysis, it was found that serum TC and HDL-C levels were inversely correlated with the risk of GC. No association was observed between serum TG levels and the risk of GC. Similarly, no association was found between serum LDL-C levels and the risk of GC.

## 1. Introduction

Gastric cancer (GC) is a type of malignant tumor that develops in the tissue of the stomach. The disease is usually caused by mutations and uncontrolled growth of cells in the inner lining of the stomach wall [1, 2]. Cancer is a major impediment to increasing life expectancy in many countries [3], and GC is among the leading illnesses contributing to the rising global burden of cancer [4]. In 2020, there were 1,089,103 new cases of GC worldwide, leading to 768,793 deaths [5]. Helicobacter pylori infection has been identified as a risk factor for GC [6], while recent research suggests a connection between blood lipid levels and GC risk [7]. In medicine, a serum lipid examination typically includes the measurement of four main elements: total cholesterol (TC), triglyceride (TG), high-density lipoprotein cholesterol (HDL-C), and low-density lipoprotein cholesterol (LDL-C) [8]. The significance of serum blood lipid levels in assessing the risk of cardiovascular disease is well established [9–11]. Moreover, research has linked blood lipid levels to the risk of developing cancer [12]. The association between blood lipid levels and the risk of colorectal [13] and lung cancer [14] has been firmly established. However, despite numerous studies investigating the association between serum lipid levels and GC risk, the findings have been inconsistent. While some studies have found no significant association between serum TC levels and GC risk [15–17], others have reported a negative association between the two variables [7, 18, 19]. Additionally, one study found this association to be present only in men [20]. Furthermore, while some studies have shown a negative link between GC risk and HDL-C [7], others have arrived at the opposite conclusion [16, 17, 21].

Conflicting evidence exists regarding the link between lipid levels and the risk of GC. Therefore, this meta-analysis incorporates all relevant prospective studies to clarify the association between lipid levels and the risk of GC.

## 2. Materials and methods

### 2.1 Eligibility criteria

The inclusion criteria were as follows: (1) cohort studies; (2) studies with adult participants (i.e., over 18 years old); (3) studies that explicitly stated that GC was the outcomes; (4) studies that clearly reported the effect size of TC, TG, HDL-C, or LDL-C; and (5) studies that reported the effect size measures as RR or HR, along with the corresponding 95% CI.

The exclusion criteria were as follows: (1) studies from which we could not collect valid data; and (2) duplicate publications.

### 2.2 Search strategy

To conduct our search, we utilized a combination of MeSH (medical subject heading) keywords and non-MeSH keywords. Two researchers independently conducted searches for relevant studies in Chinese databases (CNKI, CBM, Wanfang, and VIP) and English databases (PubMed, Embase, Web of Science, and the Cochrane Library) up to October 2020. No language limitations were applied. Our search terms included serum lipid, total cholesterol, TC,

triglyceride, TG, high-density lipoprotein cholesterol, HDL-C, low-density lipoprotein cholesterol, LDL-C, and stomach neoplasms (S1 Table). To ensure the inclusion of all relevant studies, two researchers manually reviewed additional pertinent journal articles, reviews, and other sources and carefully examined the references of the included studies.

### 2.3 Data extraction

The following data were extracted into an electronic document: the first author's name, publication year, region of study subjects, type of study, sample size, follow-up duration, and the HR between the highest and lowest serum concentrations, along with the corresponding 95% CI (S2 Table). Two researchers (SCX and YF) independently extracted data from eligible studies. In cases of disagreement, the entire team would vote to resolve the issue. When multiple risk estimates were reported in a single study, the estimate with the most comprehensive adjustment for confounding factors was selected.

### 2.4 Quality assessment

The quality assessment of each study was conducted independently by two researchers (SCX and YF) using the Newcastle–Ottawa Quality Assessment Scale (NOS) [22]. The NOS score categorizes the quality of each study into three groups based on their scores: low quality (<5), medium quality (5–7), and high quality (≥8).

### 2.5 Statistical analysis

The data were analyzed using Review Manager software (version 5.4.1) and Stata software (version 17.0). The hazard ratio (HR) and associated 95% confidence interval (95% CI) were calculated using either a fixed effects or random effects model depending on the degree of heterogeneity, which was quantified using the $I^2$ statistic. Heterogeneity was considered significant, moderate, or nonexistent when the $I^2$ value was > 50%, 30%-50%, or <30%, respectively. If $I^2$ was less than 50%, a fixed effects model was used; otherwise, a random effects model was used. Meta-regression was employed to assess the variability among subgroups, while Begg's and Egger's tests were used to investigate publication bias. Statistical significance was set at a p value below 0.05. Sensitivity analysis was conducted to assess the robustness of the results.

## 3. Results

### 3.1 Study characteristics

After conducting a systematic search, a total of 10,525 studies were retrieved. After removing 670 duplicate studies, the titles and abstracts of the remaining 9,855 studies were reviewed, and 167 studies were identified for full-text screening to assess their eligibility. Following exclusion of studies that were not related to blood lipid levels and risk of GC, a total of 7 studies were selected for inclusion in this meta-analysis. Manual searches of the reference lists of the included studies yielded 3 additional papers, bringing the total number of studies included in the meta-analysis to 10 (Fig 1). The included studies involved a total of 5,564,520 individuals and 41,408 GC occurrences, and the studies were published between 1988 and 2022. The median follow-up duration was 12.2 years. Of the 10 studies included, 5 were conducted in Asia [7, 15, 19, 20, 23], and 5 were conducted in Europe [16–18, 21, 24] (S2 Table). The studies reported NOS scores ranging from 7 (4 studies) to 8 (6 studies) (S2 Table).

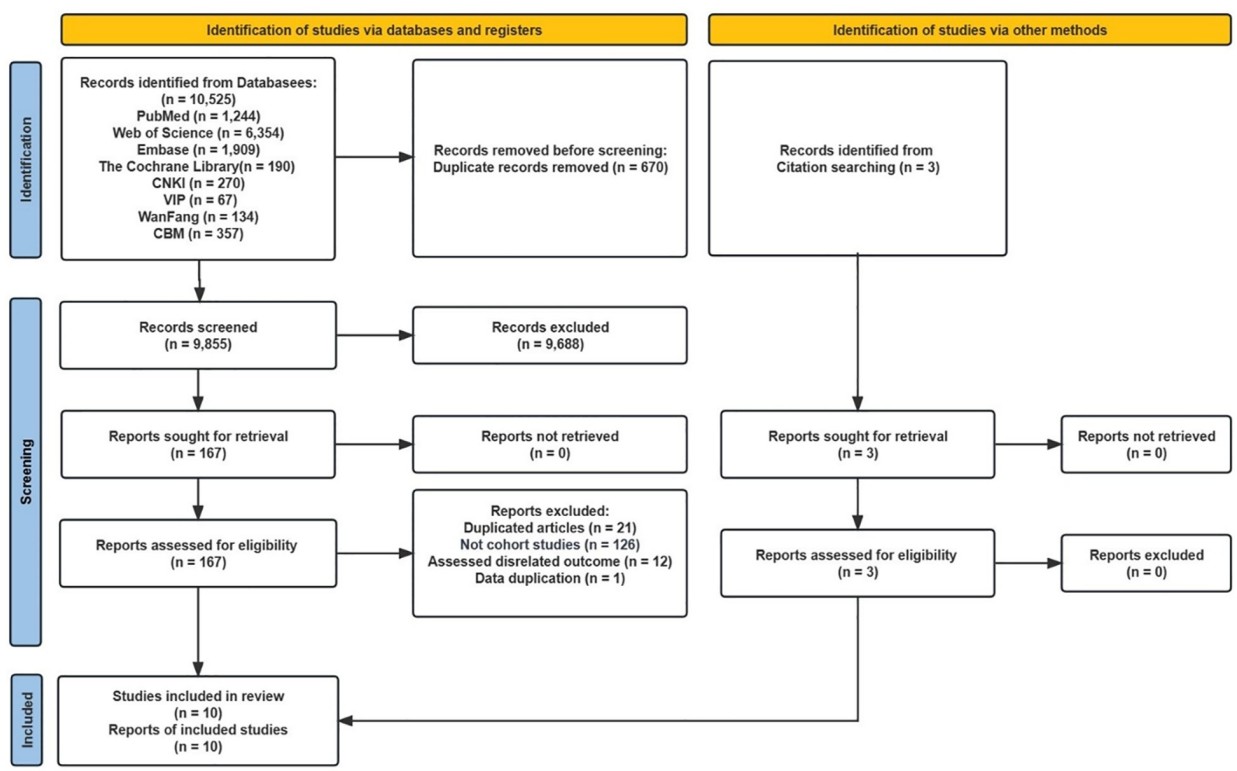

**Fig 1. Flow diagram of the studies selection process.**

## 3.2 Meta-analysis

**3.2.1 Serum total cholesterol.** A total of 8 studies published between 1988 and 2022 reported on serum TC, including 4,243,457 individuals with 40,656 GC cases. Among these studies, 5 were conducted in Asia [7, 15, 19, 20, 23], and 3 were conducted in Europe [16–18]. The results of the 8 studies showed a significant association between serum TC and GC risk (HR = 0.89, 95% CI = 0.87–0.92, P<0.01) (Fig 2). No significant heterogeneity was observed ($I^2$ = 15%, P = 0.31), and there was no evidence of publication bias based on the Egger test (P = 0.31) and Begg test (P = 0.86). The funnel plot did not exhibit any asymmetry (S1 Fig).

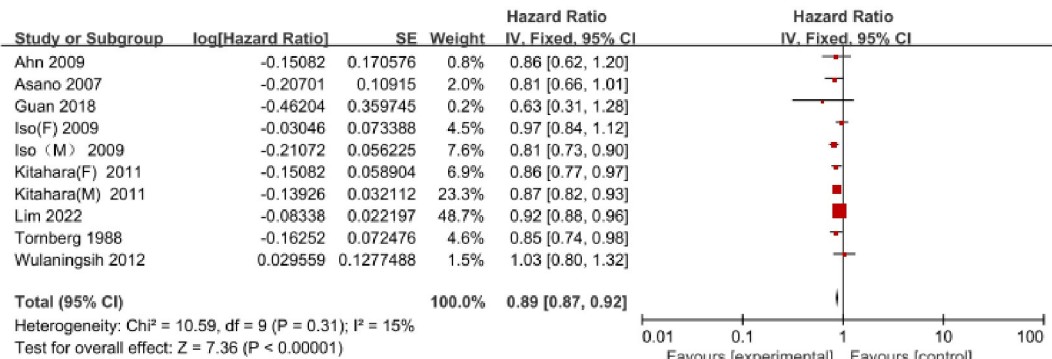

**Fig 2. Forest plot of the association between serum total cholesterol and risk of GC.**

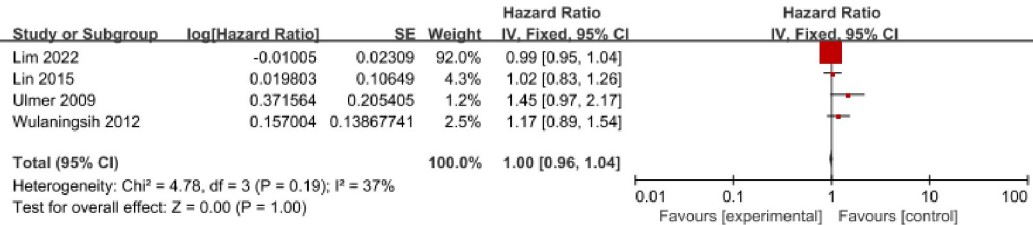

**Fig 3. Forest plot of the association between serum triglycerides and the risk of GC.**

**3.2.2 Serum triglyceride.** A total of 4 studies published between 2009 and 2022 reported on serum TG, including 3,611,979 individuals with 19,177 GC cases. Among these studies, 1 was conducted in Asia [7], and 3 were conducted in Europe [17, 21, 24]. The results of the 4 studies showed no significant association between serum TG and GC risk (HR = 1.00, 95% CI = 0.96–1.04, P = 1.00) (Fig 3). There was moderate heterogeneity ($I^2$ = 37%, P = 0.19), and no evidence of publication bias was found based on the Egger test (P = 0.12) and Begg test (P = 0.09). The funnel plot did not exhibit any asymmetry (S1 Fig).

**3.2.3 Serum high-density lipoprotein cholesterol.** A total of 4 studies published between 2009 and 2022 reported on serum HDL-C, including 3,484,919 individuals with 19,196 GC cases. Among these studies, 1 was conducted in Asia [7], and 3 were conducted in Europe [16, 17, 21]. The results of the 4 studies showed a significant association between serum HDL-C and GC risk (HR = 0.90, 95% CI = 0.86–0.93, P<0.01) (Fig 4). No heterogeneity was observed ($I^2$ = 0%, P = 0.67), and there was no evidence of publication bias based on the Egger test (P = 0.18) and Begg test (P = 0.09). The funnel plot did not exhibit any asymmetry (S1 Fig).

**3.2.4 Serum low-density lipoprotein cholesterol.** A total of 3 studies published between 2012 and 2022 reported on serum LDL-C, including 3,331,682 individuals with 18,573 GC cases. Among these studies, 2 were conducted in Asia [7, 15], and 1 was conducted in Europe [17]. The results of the 3 studies showed no significant association between serum LDL-C and GC risk (HR = 0.96, 95% CI = 0.91–1.00, P = 0.07) (Fig 5). No heterogeneity was observed ($I^2$ = 0%, P = 0.47), and there was no evidence of publication bias based on the Egger test (P = 0.31) and Begg test (P = 0.60). The funnel plot did not exhibit any asymmetry (S1 Fig).

## 3.3 Sensitivity analysis and subgroup analysis

A sensitivity analysis was conducted to examine the relationship and variability between serum levels and the risk of GC in greater depth. The findings of the study were consistent, as the results remained unaffected even when an individual study was excluded.

We conducted several targeted subgroup analyses based on the characteristics of the study to investigate the sources of heterogeneity (Table 1). However, due to the insufficient number

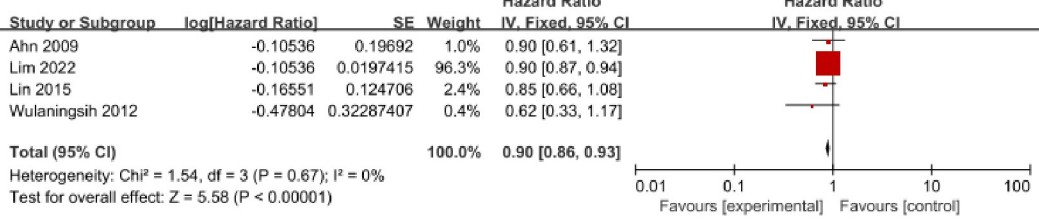

**Fig 4. Forest plot of the association between serum high-density lipoprotein cholesterol and risk of GC.**

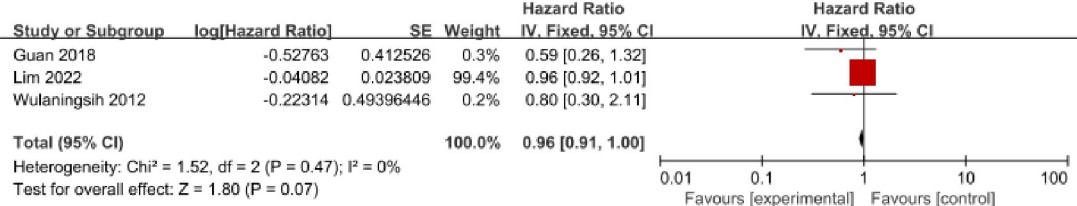

**Fig 5. Forest plot of the association between serum low-density lipoprotein cholesterol and risk of GC.**

of studies on TGs, HDL-C, and LDL-C, we only conducted a subgroup analysis of TC. The subgroups analyzed included Asia (HR = 0.88, 95% CI = 0.84–0.92), Europe (HR = 0.89, 95% CI = 0.79–1.00), follow-up duration of ≤12 years (HR = 0.92, 95% CI = 0.88–0.96), follow-up duration of >12 years (HR = 0.86, 95% CI = 0.83–0.90), high-quality studies (HR = 0.89, 95% CI = 0.86–0.93), and medium-quality studies (HR = 0.88, 95% CI = 0.80–0.97).

## 4. Discussion

This meta-analysis examined the association between blood lipid levels and the risk of GC by analyzing all prospective studies on this topic. The study included a substantial number of participants, making the results highly reliable. The findings revealed that serum TG and LDL-C levels were not associated with the risk of GC. However, higher levels of serum TC and HDL-C were associated with a lower risk of GC. The sensitivity analyses and publication bias tests supported the validity of these findings. These results help to clarify the earlier debate on

**Table 1. Subgroup analysis of the association between serum total cholesterol level and risk of gastric cancer.**

| | | Number of studies | HR (95%CI) | I² | Pª | Pᵇ |
|---|---|---|---|---|---|---|
| Areas | Asian | 5 | 0.88 (0.84,0.92) | 32.00 | 0.18 | 0.82 |
| | Europe | 3 | 0.89 (0.79,1.00) | 0 | 0.42 | |
| Duration of follow-up | ≤12 | 3 | 0.92 (0.88,0.96) | 0 | 0.39 | 0.09 |
| | >12 | 5 | 0.86 (0.83,0.90) | 0 | 0.65 | |
| Quality of study | Medium quality | 6 | 0.88 (0.80,0.97) | 26 | 0.24 | 0.99 |
| | High quality | 2 | 0.89 (0.86,0.93) | 8.2 | 0.35 | |
| **Adjustment for confounding factors** | | | | | | |
| Hyperlipidemia medication use | YES | 2 | 0.91 (0.87,0.95) | 62 | 0.07 | 0.31 |
| | NO | 6 | 0.87 (0.83,0.91) | 0 | 0.80 | |
| BMI | YES | 6 | 0.88 (0.85,0.92) | 21.10 | 0.26 | 0.65 |
| | NO | 2 | 0.91 (0.78,1.09) | 41.50 | 0.19 | |
| Exercise | YES | 4 | 0.9 (0.87,0.93) | 0 | 0.44 | 0.59 |
| | NO | 4 | 0.88 (0.8,0.95) | 33.80 | 0.20 | |
| Smoking | YES | 6 | 0.88 (0.85,0.92) | 21.10 | 0.26 | 0.50 |
| | NO | 2 | 0.91 (0.76,1.09) | 41.50 | 0.19 | |
| Alcohol drinking | YES | 5 | 0.89 (0.85,0.93) | 26.20 | 0.23 | 0.50 |
| | NO | 3 | 0.87 (0.78,0.98) | 10.50 | 0.33 | |
| Two aforementioned confounding factors | YES | 6 | 0.88 (0.85,0.92) | 21.00 | 0.26 | 0.13 |
| | NO | 2 | 0.91 0.76,1.09 | 41.50 | 0.19 | |

Pª: P values for subgroup study heterogeneity.

Pᵇ: P values of meta regression.

the association between blood lipid levels and the risk of GC. Furthermore, these findings can help with the use of blood lipid profiles to predict the occurrence of GC in high-risk populations in the future.

Previous studies have suggested a potential link between high serum TC levels and an increased risk of cancer [13]. For this meta-analysis, we systematically reviewed 8 studies [7, 15–20, 23] investigating the association between TC and GC risk, but we found conflicting results. Even subgroups based on study area, follow-up duration, and study quality showed similar outcomes. A previous meta-analysis on blood lipid levels and lung cancer risk also reported comparable findings [14]. One hypothesis for this association is that cholesterol plays a crucial role in the production of steroid hormones [25, 26], which are essential for antitumor procedures [27, 28]. For example, androgens and glucocorticoids can control antitumor immune responses by influencing the metabolism and expression of CD8 T cells in the tumor microenvironment [29]. Moreover, research indicates that hormone replacement therapy (HRT) users have a lower risk of GC [30], while those who use tamoxifen anti-hormone therapy may have an increased risk of GC [31]. However, further investigation is necessary to elucidate the underlying mechanisms.

While earlier studies [16, 17, 21] did not find any connection between HDL-C and the risk of GC, recent research [7] has shown that higher serum HDL-C levels are associated with a decreased risk of GC. In this study, the findings from various studies were integrated for the first time, revealing a negative association between HDL-C and the risk of GC with no heterogeneity ($I^2 = 0$). This association is consistent with a previous meta-analysis of prospective studies [32] and a prospective trial with 116728 samples monitored for 25 years [33], both of which reported that high serum HDL-C levels reduce the risk of gastrointestinal cancer. The underlying mechanism may be due to the antioxidant and anti-inflammatory properties of HDL-C, which can inhibit the proliferation of cancer cells. HDL-C can remove damaging oxidants, which can harm healthy DNA cells and encourage their transformation into cancer cells [34]. By preventing cell damage, HDL-C's antioxidant qualities may ultimately halt the development of cancerous cells [35]. The role of HDL in the prevention of GC was previously overlooked due to the unclear connection between HDL and the risk of GC. Further investigation of its potential benefits in decreasing the occurrence of GC should be pursued in the future.

Based on a comprehensive analysis of four papers [7, 17, 21, 24], it was found that the risk of GC was not associated with serum TG levels. The analysis revealed moderate heterogeneity ($I^2 = 37\%$) and strong stability in the sensitivity analysis. In addition, a nested case–control study conducted by Loosen [36] involving 61,936 participants also did not find any association between TGs and gastrointestinal malignancies. However, a separate study [37] found that premenopausal women with high serum TG concentrations had an increased risk of developing breast cancer. It appears that the effect of TG on cancer risk varies by site. Therefore, further research is necessary to fully explore the relationship between TG and other malignancies.

This meta-analysis revealed that there is no significant association between LDL-C level and GC risk, although some case–control studies [38, 39] have reported the opposite. One possible explanation for this discrepancy is that the low-density lipoprotein receptor (LDLR) plays a crucial role in maintaining cholesterol balance by clearing LDL-C from the bloodstream [40, 41]. Research has shown that abnormal expression of LDLR in certain malignancies can lead to abnormal serum LDL-C levels [42]. Furthermore, LDL-C plays an essential role in transporting cholesterol outside the liver [43], which may be important for the conversion of TC into steroid hormone-mediated antitumor immunity. Thus, the positive association between LDL-C and GC risk observed in some case–control studies may be a secondary phenomenon resulting from GC cell proliferation.

Our study possesses several notable strengths. This meta-analysis specifically focused on prospective cohort studies, rather than case–control studies, thus mitigating the potential limitations of case–control designs that may affect the accuracy of the results. By including a substantial number of individuals and high-quality studies, this meta-analysis provides a highly reliable analysis of the data. Moreover, the results of the sensitivity analysis indicated that the findings were robust and consistent, and there was no evidence of publication bias in any of the studies. Overall, these factors contribute to the credibility and validity of the study's conclusions.

However, importantly, this study has certain limitations that need to be considered. The effect size of the comparison between maximum and minimum serum concentrations was considered for inclusion criteria. However, differences in how these values were defined across studies may have skewed the results of our analysis. For instance, some studies divided their groups based on the interquartile range of serum concentration, while others used the normal range. Therefore, caution should be exercised when interpreting our findings. Furthermore, the studies that were analyzed herein only included populations from Asia and Europe, which limits the overall representativeness of the analysis's findings.

## 5. Conclusion

In summary, this meta-analysis found no significant relationship between the risk of GC and serum levels of TG and LDL-C. However, there was an inverse association between GC risk and serum levels of TC and HDL-C. Further investigation is necessary to assess the efficacy of clinical therapies in reducing the incidence of GC, as well as to fully understand the underlying mechanisms of action.

## Supporting information

**S1 Checklist. PRISMA 2020 checklist.**
(PDF)

**S1 Table. Searching strategy for PubMed.**
(PDF)

**S2 Table. Characteristics of the included studies.**
(PDF)

**S1 Fig. Funnel plot of the association between lipid levels and the risk of gastric cancer.**
(RAR)

## Author Contributions

**Conceptualization:** Shicong Xu, Xianrong Li.

**Data curation:** Shicong Xu, Ying Fan, Ling Zhang.

**Formal analysis:** Shicong Xu, Yuyue Tan.

**Software:** Ling Zhang.

**Writing – original draft:** Shicong Xu.

**Writing – review & editing:** Shicong Xu.

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
