## [Decision Letter · Decision Letter 0]

18 Apr 2023

PONE-D-23-03785Association between blood lipid levels and risk of gastric cancer: a systematic review and meta-analysisPLOS ONE

Dear Dr. Xianrong,

Thank you for submitting your manuscript to PLOS ONE. After careful consideration, we feel that it has merit but does not fully meet PLOS ONE’s publication criteria as it currently stands. Therefore, we invite you to submit a revised version of the manuscript that addresses the points raised during the review process. Please insert quality assessment score in supplementary Table-S2.-Update PRISMA flowchart in accordance with PRISMA 2020

We look forward to receiving your revised manuscript.

Kind regards,

Ozra Tabatabaei-Malazy

Academic Editor

PLOS ONE

Journal Requirements:

Reviewers' comments:

Reviewer's Responses to Questions

**Comments to the Author**

1. Is the manuscript technically sound, and do the data support the conclusions?

Reviewer #1: Yes

Reviewer #2: Yes

2. Has the statistical analysis been performed appropriately and rigorously? 

Reviewer #1: Yes

Reviewer #2: Yes

3. Have the authors made all data underlying the findings in their manuscript fully available?

Reviewer #1: Yes

Reviewer #2: Yes

4. Is the manuscript presented in an intelligible fashion and written in standard English?

Reviewer #1: Yes

Reviewer #2: No

5. Review Comments to the Author

Reviewer #1: Dear Editor

I have read the in which, in a systematic review authors tried to estimate the risk of abnormal levels of components of lipid profile on gastric cancer. The study is well designed and authors tried their best to precisely define the question and include the studies that answer their question. I believe that the study merit publication, however, I have some concerns that need to be answered

1- Usually subjects with abnormal lipid profile receive medications which may or may not affect the occurrence of final outcome which is gastric cancer. What measures or strategies authors used to overcome such bias?

2- In some of included studies the RR was related to risk highest levels of measured lipid (for example TC) compare to lowest levels (Ahn et al 2009), while in some other studies only the risk of levels was evaluated. How the authors adjusted this?

3- there should be a table presenting the characteristics of studies, including the year of publication, number of subjects in each group, duration, ….

4- there are some grammatical errors. The manuscript needs to be revised grammatically.

Reviewer #2: In this manuscript, Shicong and colleagues conducted a systematic review and meta-analysis to assess the association between blood lipid levels and risk of gastric cancer. The methodology is appropriate. However, the manuscript requires thorough language editing.

Following are my comments:

1- There should be no references in the abstract.

2- GC has not been defined on its appearance in the introduction.

3- I believe a Table containing the summary findings of each included studies should be included in the main text.

4- I could not find the supplemental data so I was not able to assess the manuscript fully.

6. PLOS authors have the option to publish the peer review history of their article (what does this mean?). If published, this will include your full peer review and any attached files.

Reviewer #1: No

Reviewer #2: No

---

## [Author Response · Author response to Decision Letter 0]

14 May 2023

Response to Academic Editor:

1.Please insert quality assessment score in supplementary Table-S2.

-Update PRISMA flowchart in accordance with PRISMA 2020

Response: Thank you for your feedback regarding our manuscript. We have carefully considered your suggestions and made the necessary revisions.

To incorporate the quality assessment of the literature, we have merged it into the original Table S2 and generated a new document, which we have renamed as S2 Table. 

Additionally, we have updated the PRISMA flowchart in accordance with the PRISMA 2020 guidelines as per your recommendation.

Thank you again for your valuable input, and please let us know if you have any further suggestions or comments.

Response: Thank you for your feedback. We apologize for any inconvenience caused by not adhering to PLOS ONE's style requirements, including those for file naming. We have revised our manuscript to ensure that it meets the specified guidelines. If you have any further suggestions or comments, please let us know. We appreciate your time and attention to our submission.

3.In your Data Availability statement, you have not specified where the minimal data set underlying the results described in your manuscript can be found. PLOS defines a study's minimal data set as the underlying data used to reach the conclusions drawn in the manuscript and any additional data required to replicate the reported study findings in their entirety. All PLOS journals require that the minimal data set be made fully available. For more information about our data policy, please see http://journals.plos.org/plosone/s/data-availability.

Response: Thank you for your feedback on our manuscript. We appreciate the opportunity to clarify our data availability statement. We understand that PLOS defines the minimal data set as the underlying data used to reach the conclusions in the manuscript and any additional data required to replicate the reported study findings in their entirety. We apologize for the oversight in not specifying where the minimal data set can be found.

We have updated the data availability statement to include the location of the minimal data set. The minimal data set can now be found at Supporting Information. Furthermore, it should be noted that the studies included in our research have been cited, and readers can easily access the corresponding data within these studies. We have ensured that the data set is fully available and can be accessed by readers to replicate our findings.

Thank you for bringing this to our attention and please let us know if you have any further concerns or suggestions.

Response: Thank you for bringing to my attention the requirement for the corresponding author to have an ORCID iD in Editorial Manager for papers submitted. I have created a new ORCID iD and have validated it in Editorial Manager as per the instructions provided. To link my ORCID iD to my Editorial Manager account, I followed the steps outlined in the video tutorial provided and successfully completed the process. Thank you for your assistance in ensuring that my submission meets the necessary requirements.

5.Please include captions for your Supporting Information files at the end of your manuscript, and update any in-text citations to match accordingly. Please see our Supporting Information guidelines for more information: http://journals.plos.org/plosone/s/supporting-information.

Response: Thank you for your valuable feedback. I have carefully reviewed your suggestion and have made the necessary changes by including captions for all the Supporting Information files at the end of the manuscript.

I appreciate your guidance and have followed the guidelines provided in the Supporting Information guidelines for PLOS ONE. Please let me know if there are any further changes or revisions that you would like me to make.

Response to Reviewer #1：

1- Usually subjects with abnormal lipid profile receive medications which may or may not affect the occurrence of final outcome which is gastric cancer. What measures or strategies authors used to overcome such bias?

Response: We appreciate your concern regarding potential biases introduced by the use of lipid-lowering medications among study participants.

To address the potential bias, we conducted subgroup analyses and a meta-regression to investigate the impact of lipid-lowering medication usage, which are presented in Table 1. Additionally, we performed sensitivity analyses. We believe these measures effectively controlled for the influence of lipid-lowering medication usage on the study findings, ensuring the reliability and validity of our research.

2- In some of included studies the RR was related to risk highest levels of measured lipid (for example TC) compare to lowest levels (Ahn et al 2009), while in some other studies only the risk of levels was evaluated. How the authors adjusted this?

Response: Thank you for your feedback. In response to the concern about the correction of blood lipid levels in our study, we would like to clarify that we carefully evaluated the methods and results of each study to determine appropriate adjustments. When extracting effect sizes, we obtained hazard ratios (HR) or relative risks (RR) with the most comprehensive adjustment for confounding factors and their corresponding 95% confidence intervals. In addition, we performed subgroup analyses for confounding factors that were adjusted in the included studies. We believe that our measures adequately addressed potential confounding effects and provided a more accurate estimate of the association between blood lipid levels and the risk of the outcome of interest.

3- there should be a table presenting the characteristics of studies, including the year of publication, number of subjects in each group, duration, ….

Response: Thank you for your feedback. In response to the suggestion regarding the presentation of the characteristics of the included studies, we have created a separate document named "S2 Table" to provide a comprehensive overview of the year of publication, number of subjects in each group, duration, and other relevant information for each study.

We apologize for any confusion caused by our initial placement of this information in the supplementary materials as Table S2. We hope that this revised approach will make it easier for readers to access and evaluate the characteristics of the included studies.

4- there are some grammatical errors. The manuscript needs to be revised grammatically.

Response: Thank you for reviewing our manuscript and for providing feedback on the grammatical errors. We apologize for any inconvenience caused by the grammatical errors in our initial submission.

We have thoroughly revised the manuscript to correct all of the grammatical errors. We have carefully reviewed and edited the text to ensure that it meets high standards of grammar and style. Additionally, we had our manuscript reviewed by a professional editor to ensure its quality.

Once again, we thank you for bringing this to our attention and we appreciate your valuable feedback. Please let us know if you have any further suggestions or comments.

Response to Reviewer #2：

1- There should be no references in the abstract.

Response: Thank you for your feedback. We appreciate your suggestion regarding the references in the abstract. We have carefully reviewed our manuscript and removed all references from the abstract to comply with the journal's guidelines.

2- GC has not been defined on its appearance in the introduction.

Response: Thank you for your feedback. We have carefully reviewed our manuscript and have made the necessary revisions to define the term GC in the introduction. We have highlighted the revised text in yellow for your convenience.

3- I believe a Table containing the summary findings of each included studies should be included in the main text.

Response: Thank you for your feedback. We appreciate your suggestion regarding the inclusion of a summary table for the findings of each included study in the main text. We would like to clarify that we have already addressed this concern in our previous correspondence. We have created a separate document named "S2 Table" to provide a comprehensive overview of the characteristics and findings of each included study. This table will be included in the resubmission of our manuscript. We apologize for any confusion caused by the initial placement of this information in Table-S2 in our initial submission. We hope that the revised approach will make it easier for readers to access and evaluate the findings of the included studies.

4- I could not find the supplemental data so I was not able to assess the manuscript fully.

Response: Thank you for bringing to our attention the issue regarding the supplemental data. We apologize for any inconvenience caused by the inability to locate the data in our initial submission. We have carefully reviewed and revised the supplemental data according to PLOS ONE guidelines and have reorganized it to make it easily accessible in the revised submission. We have also provided clear instructions for locating the supplemental data in the cover letter. We hope that this revised approach will enable you to access and assess the supplemental data fully. We appreciate your valuable feedback, and we remain committed to ensuring that all components of our manuscript are presented accurately and transparently.

---

## [Decision Letter · Decision Letter 1]

29 May 2023

PONE-D-23-03785R1Association between blood lipid levels and risk of gastric cancer: a systematic review and meta-analysisPLOS ONE

Dear Dr. Xianrong,

Thank you for submitting your manuscript to PLOS ONE. After careful consideration, we feel that it has merit but does not fully meet PLOS ONE’s publication criteria as it currently stands. Therefore, we invite you to submit a revised version of the manuscript that addresses the points raised during the review process.

ACADEMIC EDITOR:

Please re-upload PRISMA flowchart in accordance with the attached file. Moreover, consider minor English language editing in whole manuscript.

We look forward to receiving your revised manuscript.

Kind regards,

Ozra Tabatabaei-Malazy

Academic Editor

PLOS ONE

Journal Requirements:

Additional Editor Comments:

Dear Authors,

Please re-upload PRISMA flowchart in accordance with the attached file. Moreover, consider minor English language editing in whole manuscript.

Reviewers' comments:

Reviewer's Responses to Questions

**Comments to the Author**

1. If the authors have adequately addressed your comments raised in a previous round of review and you feel that this manuscript is now acceptable for publication, you may indicate that here to bypass the “Comments to the Author” section, enter your conflict of interest statement in the “Confidential to Editor” section, and submit your "Accept" recommendation.

Reviewer #1: All comments have been addressed

Reviewer #2: All comments have been addressed

2. Is the manuscript technically sound, and do the data support the conclusions?

Reviewer #1: Yes

Reviewer #2: Yes

3. Has the statistical analysis been performed appropriately and rigorously? 

Reviewer #1: Yes

Reviewer #2: Yes

4. Have the authors made all data underlying the findings in their manuscript fully available?

Reviewer #1: Yes

Reviewer #2: Yes

5. Is the manuscript presented in an intelligible fashion and written in standard English?

Reviewer #1: Yes

Reviewer #2: Yes

6. Review Comments to the Author

Reviewer #1: the previous comments are responded by authors. I have no further comments. however, the manuscript needs to be corrected in regards to spelling and some minor grammatical errors.

Reviewer #2: I thank the authors for their detailed revision. All my comments have been addressed and I believe the manuscript can be considered for potential publication.

7. PLOS authors have the option to publish the peer review history of their article (what does this mean?). If published, this will include your full peer review and any attached files.

Reviewer #1: No

Reviewer #2: No

---

## [Author Response · Author response to Decision Letter 1]

9 Jun 2023

Response to Academic Editor:

1.Please re-upload PRISMA flowchart in accordance with the attached file. Moreover, consider minor English language editing in whole manuscript.

Response: Thank you for your valuable feedback. We appreciate your suggestion. We have made the necessary changes and have re-uploaded the PRISMA flowchart in accordance with the attached file. Furthermore, we have carefully reviewed the entire manuscript and have made minor English language edits where necessary. Thank you for bringing this to our attention, and we believe the revised version now meets the required standards.

2.Please review your reference list to ensure that it is complete and correct. If you have cited papers that have been retracted, please include the rationale for doing so in the manuscript text, or remove these references and replace them with relevant current references. Any changes to the reference list should be mentioned in the rebuttal letter that accompanies your revised manuscript. If you need to cite a retracted article, indicate the article’s retracted status in the References list and also include a citation and full reference for the retraction notice.

Response: Thank you for your valuable feedback and guidance on our manuscript. We have thoroughly reviewed the reference list and can confirm that it is complete and correct. We have carefully checked each reference and have not found any errors or retractions in the cited papers.

Based on our thorough examination, we are confident that our reference list accurately represents the relevant and current literature in our field of study. We have taken great care to ensure the integrity of our citations, and we believe that the references provided in our manuscript are reliable and appropriate for supporting our research.

We appreciate your diligence in highlighting the importance of maintaining an accurate and up-to-date reference list, and we are grateful for the opportunity to address this aspect of our manuscript. Should you have any further concerns or suggestions, please do not hesitate to let us know.

Thank you once again for your thorough review, and we look forward to your further evaluation of our revised manuscript.

Response to Reviewer #1：

Response: Thank you for reviewing our manuscript and for your previous comments. We appreciate your acknowledgement that the previous comments have been addressed by the authors.

We have carefully reviewed the manuscript once again, focusing specifically on spelling and minor grammatical errors. We have made the necessary corrections to ensure the accuracy and clarity of the text. Additionally, we have conducted a thorough proofreading to improve the overall language quality.

We sincerely appreciate your attention to detail and your efforts in helping us enhance the manuscript's readability. Your input has been invaluable in improving the overall quality of our work.

If you have any further comments or suggestions, please do not hesitate to let us know.

Response to Reviewer #2：

I thank the authors for their detailed revision. All my comments have been addressed and I believe the manuscript can be considered for potential publication.

Response: Thank you for your thorough review of our revised manuscript. We sincerely appreciate your time and effort in providing constructive feedback and suggestions. We are pleased to hear that you acknowledge our efforts in addressing all of your comments and that you believe the manuscript is now suitable for potential publication.

Your positive evaluation of our revised manuscript is encouraging and greatly appreciated. We are grateful for the opportunity to have our work considered for publication in PLOS ONE. We would like to express our gratitude for your valuable input, which has undoubtedly contributed to the improvement of our research.

Should there be any further steps or requirements in the publication process, please kindly let us know, and we will promptly address them.

Once again, thank you for your time and support. We look forward to the possibility of our manuscript being published in PLOS ONE.

---

## [Editor Report · Decision Letter 2]

13 Jun 2023

PONE-D-23-03785R2Association between blood lipid levels and risk of gastric cancer: a systematic review and meta-analysisPLOS ONE

Dear Dr. Xianrong,

Thank you for submitting your manuscript to PLOS ONE. After careful consideration, we feel that it has merit but does not fully meet PLOS ONE’s publication criteria as it currently stands. Therefore, we invite you to submit a revised version of the manuscript that addresses the points raised during the review process.

ACADEMIC EDITOR:

Please insert the correct format of the Fig-1 that observed in revised manuscript with track changes file.

We look forward to receiving your revised manuscript.

Kind regards,

Ozra Tabatabaei-Malazy

Academic Editor

PLOS ONE
---

## [Author Response · Author response to Decision Letter 2]

17 Jun 2023

Please insert the correct format of the Fig-1 that observed in revised manuscript with track changes file.

Response: Thank you for your feedback on my PRISMA flowchart. I have thoroughly reviewed and revised the flowchart, taking into consideration the format used in previously published studies in PLOS ONE. The flowchart has now been modified to align with the appropriate format. I appreciate your guidance in ensuring the accuracy and adherence to the journal's standards. Please let me know if there are any further revisions or modifications you would like me to address.

---

## [Editor Report · Decision Letter 3]

21 Jun 2023

Association between blood lipid levels and risk of gastric cancer: a systematic review and meta-analysis

PONE-D-23-03785R3

Dear Dr. Xianrong,

We’re pleased to inform you that your manuscript has been judged scientifically suitable for publication and will be formally accepted for publication once it meets all outstanding technical requirements.

Kind regards,

Ozra Tabatabaei-Malazy

Academic Editor

PLOS ONE
---

## [Editor Report · Acceptance letter]

27 Jun 2023

PONE-D-23-03785R3 

Association between blood lipid levels and risk of gastric cancer: A systematic review and meta-analysis 

Dear Dr. Li:

I'm pleased to inform you that your manuscript has been deemed suitable for publication in PLOS ONE. Congratulations! Your manuscript is now with our production department. 

Kind regards, 

on behalf of

Dr. Ozra Tabatabaei-Malazy 

Academic Editor

PLOS ONE